# Prevalence of malaria and associated factors among symptomatic pregnant women attending antenatal care at three health centers in north-west Ethiopia

**Andargachew Almaw** [1]*, **Mulat Yimer**[2], **Megbaru Alemu**[2], **Banchamlak Tegegne**[3]

**1** Department of Medical Laboratory Science, College of Medicine and Health Sciences, Debre Tabor University, Debre Tabor, Ethiopia, **2** Department of Medical Laboratory Science, College of Medicine and Health Sciences, Bahir Dar University, Bahir Dar, Ethiopia, **3** Amhara Public Health Institute, Bahir Dar, Ethiopia

* andargachewalmaw@gmail.com

## Abstract

### Background

Malaria is the disease caused by *Plasmodium* species and primarily transmitted by the bite of female *Anopheles* mosquitoes. During pregnancy, malaria causes life threatening outcomes to the mother, the fetus and the new born. Even though, malaria symptomatic pregnant women highly attract mosquitoes and have higher potential of transmitting the disease in communities, most of the previous studies focused on pregnant women with asymptomatic *Plasmodium* infections. Therefore, the aim of this study was to assess the prevalence of malaria and associated factors among symptomatic pregnant women attending antenatal care at three health centers in northwest Ethiopia.

### Methods

A health facility based cross-sectional study was conducted from February to April, 2021. A total of 312 malaria symptomatic pregnant women were involved from three health centers and enrolled by convenient sampling technique. A questionnaire was used to collect socio demographic and clinical data through face to face interview. Capillary blood samples were collected and used to prepare thin and thick blood smears, which were then stained using 10% Giemsa and examined under light microscope. Logistic regression was used to assess factors associated with malaria. Adjusted odds ratio with 95% confidence interval was calculated and P-value < 0.05 was considered statistically significant.

### Results

The prevalence of malaria among symptomatic pregnant women was 20.8% (65/312) of which 12.2% (38/312), 4.8% (15/312) and 3.8% (12/312) were *P. falciparum*, *P. vivax* and mixed infections, respectively. Being illiterate (p< 0.001), first trimester (p = 0.036), primigravidae (p<0.001), living far from health center (p<0.001), not sleeping under long lasting

**Data Availability Statement:** All relevant data are within the paper and its Supporting Information files.

**Funding:** The authors received no specific funding for this work.

**Competing interests:** he authors have declared that no competing interests exist.

insecticide treated nets (p<0.001) and living near irrigation areas (p = 0.006) were significantly associated with prevalence of malaria in malaria symptomatic pregnant women.

## Conclusions

Even though prevalence of malaria is decreasing in the country because of scale-up of intervention and prevention measures, this study showed that, malaria is still the major public health problem among pregnant women. Being illiterate, first trimester, primigravidae, living far from health centers, not sleeping under long lasting insecticide treated nets and living near irrigation areas were factors that increased the prevalence of malaria in malaria symptomatic pregnant women. Therefore, special attention should be given to pregnant women prone to these factors.

## Background

Malaria is the disease caused by intracellular parasites of five species of the genus *Plasmodium* namely; *Plasmodium falciparum* (*P. falciparum*), *Plasmodium vivax* (*P. vivax*), *Plasmodium ovale* (*P. ovale*), *Plasmodium malariae* (*P. malariae*) and *Plasmodium Knowlesi* (*P. knowlesi*) [1]. According to the World Health Organization (WHO) 2020 report, malaria is endemic in 87 countries in the world with 229 million cases and 409,000 deaths in 2019 [2]. Of this global burden, 215 million cases and 384,000 deaths occurred in Africa [2]. Malaria is transmitted to humans by the bite of female *Anopheles* mosquitoes. *Anopheles* mosquitoes feed on blood for survival and production of eggs. During feeding, infected female *Anopheles* mosquito inoculates the infective sporozoites from its salivary gland into human circulation. After inoculation, parasites circulate with blood and those reaching to the liver undergo one cycle development. Then, parasites infect and multiply inside red blood cells to bring the characteristic signs and symptoms [3]. The clinical manifestations include fever, joint pain, chills, headache and vomiting which usually appear between 10 and 15 days after the *Anopheles* mosquito bite. In addition, malaria may also be transmitted through blood transfusion and congenitally [4].

In Ethiopia, *P. falciparum* and *P. vivax* are the two dominant species causing malaria and accounted for 60% and 40%, respectively. *Anopheles arabiensis* is the main malaria vector; *An. pharoensis*, *An. funestus* and *An. nili* play a role as secondary vectors [5].

Malaria diagnosis is done using microscopic examination, rapid diagnostic test and molecular techniques. The detection of *Plasmodium* parasites by light microscopy from capillary or venous blood is still the primary method in most health facilities across the world [6]. Early diagnosis and treatment play an important role in malaria prevention and control. Artemisinin based combination therapy and chloroquine are first-line anti-malaria treatments for *P. falciparum* and *P. vivax*, respectively. Oral quinine is used as the first-line treatment for pregnant women during the first trimester [5]. Vector control using long lasting insecticide treated nets, indoor residual spraying and larval source management are used for prevention of malaria in Ethiopia [5].

Malaria affects both sexes and all age groups. However, infection during pregnancy poses severe and life threatening outcomes to the mother, the fetus, and the new born. Because, during pregnancy, there is decline in immunity and sequestration of infected erythrocytes to the placenta [7, 8]. Maternal complications of malaria in pregnancy include; anemia, cerebral malaria and death [4]. For instance, malaria related anemia is estimated to cause 10,000 maternal deaths each year in Africa [9]. Major complications among fetuses and newborns include;

low birth weight, intrauterine growth retardation, abortion, premature delivery and fetal death [10]. In sub-Saharan Africa, among the 11 million pregnant women exposed to malaria in 2018, about 872,000 newborns were born with low birth weight [11]. Although, pregnant women have higher potential of transmitting malaria than their counter parts [12], community chemotherapy campaigns often exclude them. So, they may contribute to persistent malaria transmission in the area.

The prevalence of malaria among asymptomatic pregnant women has been well documented in Ethiopia [13–16]. However, prevalence among symptomatic pregnant women remains poorly considered. So, there is scarcity of data on prevalence of malaria and associated factors among symptomatic pregnant women. In addition, Ethiopia has set goals for malaria elimination and conducting confirmatory testing for 100% of suspected malaria cases and treat as per the guide line is one of the main strategic objectives [5]. Therefore, this study aimed to assess the prevalence of malaria and associated factors among symptomatic pregnant women attending antenatal care at three health centers in northwest Ethiopia.

## Materials and methods

### Study design, period and area

A health facility based cross-sectional study was conducted at three health centers (Tis Abay, Zenzelma and Hamusit) in northwest Ethiopia from February to April, 2021. The three health centers were selected purposively due to logistic reasons. Zenzelma and Hamusit are located at 15km and 40 km, respectively North of Bahir Dar city. Tis Abay is located at 35km east of Bahir Dar city. The climate of all the three areas is Woyna dega (1500m-2500m). The Altitude of Hamusit, Zenzelma and Tis Abay areas is 2077 meters, 1800 meters and 1653 meters, respectively above sea level. The three study areas have similar climatic condition with main rainy season occurring from June to August. Similarly, the study areas have an intensive irrigation system used for cultivation of rice, onion, sugarcane etc. Malaria transmission occurs throughout the year in the areas but the peak incidence and epidemics occur seasonally at the end of September and beginning of May during planting and harvesting seasons. About 43,128, 11,650 and 55,426 number of catchment populations get health service from Tis Abay, Zenzelma and Hamusit health centers, respectively. There were 336, 90 and 776 pregnant women diagnosed for malaria from Tis Abay, Zenzelma and Hamusit health centers, respectively in 2019/2020.

### Sample size determination and sampling technique

Using single population proportion formula with 50% prevalence, 95% confidence level, 5% margin of error and 10% non-response rate the total sample size was 422. However, the total number of symptomatic pregnant women in the study areas was 1202 based on 2019/2020 data from all health centers which is finite population or less than 10.000. So, adjustment (correction) formula was used and 312 malaria symptomatic pregnant women have participated in the study. Finally; using proportionate allocation, 87, 24 and 201 study participants were involved from Tis Abay, Zenzelma and Hamusit health centers, respectively and enrolled by convenient sampling technique.

### Dependent and independent variables

The dependent variable was malaria infection status. The independent variables were age, residence, educational status, occupation, gestational age, gravidity, LLINs use, distance from health centers and Irrigation.

## Operational definitions

Symptomatic: Women who exhibited at least one of the signs and/or symptoms of malaria like fever (axillary temperature $\geq37.5°C$), joint pain, malaise, vomiting, chills etc.

Pregnant: Women of confirmed urine Human Chorionic Gonadotropin (HCG) hormone positive in laboratory.

Distance was estimated as:

Far = Distance from health centers to pregnant women's home covering 2 hours and more on foot

Medium = Distance from health centers to pregnant women's home covering 1–2 hrs on foot

Near = Distance from health centers to pregnant women, s home cover less than 1 hour on foot

## Data collection and processing

**Clinical and socio-demographic data collection.** Women confirmed of being pregnant by using Human Chorionic Gonadotropin (HCG) hormone test from urine sample in laboratory attend their follow up. So, pregnant women exhibiting signs and/or symptoms of malaria during their visit were asked for willingness to participate in the study. From volunteered pregnant women, clinical and socio- demographic data were collected through face to face interview by the midwives using a structured questionnaire.

**Blood sample collection.** For detection and identification of *Plasmodium* species, capillary blood was collected by trained and experienced medical laboratory technicians from each health center and the principal investigator. Pregnant women's finger was cleaned with 70% ethyl alcohol and the side of fingertip was pricked with a sterile lancet. The first drop of blood which contains tissue fluids was wiped away. One μl and 2 μl of blood were used for preparation of thin and thick blood films, respectively. The prepared blood films were air dried and thin films were fixed with absolute methanol. The smears were then stained by 10% giemsa stain and examined under light microscope following standard operating procedures. A negative result was reported after checking at least 100 oil immersion fields. Thick blood films were used for parasite detection and thin blood films were used for species identification.

## Data quality control

Before starting data collection, training was given to data collectors (Midwives and Medical Laboratory Technicians) by principal investigator on how to collect socio-demographic and clinical data and process laboratory data. To ensure the quality of Giemsa stain, a quality giemsa stock solution prepared in Amhara public health institute (APHI) was used for preparation of 10% Giemsa (working solution) and prepared every 8 hours. Buffered water with PH- value of 7.2 was used for preparation of Giemsa stain working solution and filtered before use. At the end of data collection, all the slides were re-examined by experienced malaria microscopists in APHI who have not been engaged during data collection. This review (re-examination) was taken as final.

## Data analysis

Questionnaire containing socio-demographic characteristics, clinical data and associated factors was checked for completeness. The data were coded, entered, cleaned and analyzed using Statistical Package for Social Sciences version 20 (SPSS 20).

Descriptive statistics (frequencies, mean and percentage) was used to explain the study participants in relation to the variables. Logistic regression was used to determine factors associated with malaria. Adjusted Odds ratios (AORs) with 95% confidence interval were calculated and P- value < 0.05 was considered statistically significant.

### Ethical considerations

Ethical approval was obtained from College of Medicine and Health Sciences, Bahir Dar University, Institutional Review Board (protocol number 157/2021). Support letter was also obtained from APHI, South Gondar and Bahir Dar city administration Zonal health departments, Dera district health office and the three Health centers. Written informed consent was obtained from study participants after explaining the purpose of the study by data collectors. Study participants with positive results were treated according to the national malaria treatment guideline.

## Results

### Sociodemographic characteristics of study participants

Overall, 312 malaria symptomatic pregnant women comprising of 87.8% (274/312) rural and 12.2% (38/312) urban dwellers participated in the study. Of these, 58% (181/312) were illiterate, 70.2% (219/312) were in the age group of 21–30 years old and 81.1% (253/312) were farmers (Table 1).

### Prevalence of malaria among symptomatic pregnant women

The overall prevalence of malaria was 20.8% (65/312) (95% CI: 16.7%-25.6%). The highest relative proportion of malaria was found among 1st trimester of pregnancy, 44.6% (29/65). In relation to gravidity, the highest relative proportion of malaria was detected among primigravidae, 67.7% (44/65) (Table 2).

### Factors associated with malaria among symptomatic pregnant women

Illiterate Pregnant women were more affected than those who had attended secondary school and above (AOR = 7.9; 95%CI: 2.55,-24.50, p<0.001). Pregnant women in 1st trimester and primigravidae had 2.6 and 8.3 times increased susceptibility to malaria infection than pregnant women of 3rd trimester and multigravidae, respectively. However, age and residence of pregnant women were not statistically associated with malaria infection (Table 3).

**Table 1. Sociodemographic characteristics of study participants attending ANC at three health centers in northwest Ethiopia, February to April 2021.**

| Variables | Category | Frequency (%) |
|---|---|---|
| Age group | ≤20 | 43 (13.8) |
| | 21–30 | 219 (70.2) |
| | 31–40 | 50 (16.0) |
| Residence | Rural | 274 (87.8) |
| | Urban | 38 (12.2) |
| Educational status | Illiterate | 181 (58.0) |
| | Primary | 68 (21.8) |
| | Secondary and above | 63 (20.2) |
| Occupation | Farmer | 253 (81.1) |
| | Private business | 43 (13.8) |
| | Employed | 16 (5.1) |

**Table 2. Prevalence of malaria among symptomatic pregnant women attending ANC at three health centers in northwest Ethiopia from February to April 2021 (n = 312).**

| Result | | Proportion % |
|---|---|---|
| Total positive | | 20.8% (65/312) (95% CI: 16.7–25.6) |
| *P. falciparum* | | 12.2% (38/312) (95% CI: 9–16.2) |
| *P. vivax* | | 4.8% (15/312) (95% CI: 2.9–7.8) |
| Mixed (*P. falciparum & P. vivax*) | | 3.8% (12/312) (95% CI: 2.2–6.6) |
| Gestational age | First trimester | 44.6% (29/65) |
| | Second trimester | 32.3% (21/65) |
| | Third trimester | 23.1% (15/65) |
| Gravidity | Primigravidae | 67.7% (44/65) |
| | Secundigravidae | 10.8% (7/65) |
| | Multigravidae | 21.5% (14/65) |

## Discussion

Malaria remains a leading cause of morbidity and mortality especially among pregnant women and children in the developing world [17]. In the present study, the prevalence of malaria was 20.8% which is in line with the prevalence in Ghana (22%) [18]. However, the

**Table 3. Factors associated with malaria among symptomatic pregnant women attending ANC at three health centers in northwest Ethiopia, February to April 2021.**

| Variables | Category | Malaria | | COR (95%CI) | P-value | AOR (95%CI) | P-value |
|---|---|---|---|---|---|---|---|
| | | Positive | Negative | | | | |
| Age (Yrs.) | ≤20 | 13 | 30 | 3.9 (1.26, 12.08) | 0.018* | 1.5 (0.33,7.05) | 0.589 |
| | 21–30 | 47 | 172 | 2.5 (0.92, 6.54) | 0.072 | 1.7 (0.51,5.57) | 0.398 |
| | 31–40 | 5 | 45 | 1 | | | |
| Residence | Rural | 62 | 212 | 3.4 (1.02, 11.47) | 0.047* | 0.6 (0.13,2.48) | 0.442 |
| | Urban | 3 | 35 | 1 | | | |
| Educational status | Illiterate | 51 | 130 | 3.7 (1.51, 9.18) | 0.004* | 7.9 (2.55,24.50) | < 0.001* |
| | Primary | 8 | 60 | 1.3 (0.41, 3.88) | 0.679 | 0.9 (0.25,3.04) | 0.818 |
| | secondary and above | 6 | 57 | 1 | | | |
| Gestational age | 1st trimester | 29 | 67 | 2.6 (1.3, 5.17) | 0.008* | 2.6 (1.07, 6.17) | 0.036* |
| | 2nd trimester | 21 | 91 | 1.4 (0.66,2.83) | 0.395 | 1.4 (0.55, 3.43) | 0.497 |
| | 3rd trimester | 15 | 89 | 1 | | | |
| Gravidity | Primigravidae | 44 | 90 | 4.0 (2.04, 7.65) | < 0.001* | 8.3 (3.11, 22.25) | < 0.001* |
| | Secondi gravidae | 7 | 44 | 1.3 (0.49, 3.39) | 0.614 | 1.9 (0.57, 6.02) | 0.307 |
| | Multigravidae | 14 | 113 | 1 | | | |
| Distance from HCs to home | Far | 20 | 20 | 5.1 (2.43, 10.55) | < 0.001* | 2.7 (1.00, 7.26) | < 0.001* |
| | Medium | 15 | 75 | 1.0 (0.51, 2.00) | 0.969 | 0.9 (0.41, 2.14) | 0.883 |
| | Near | 30 | 152 | 1 | | | |
| Sleeping in LLINs | No | 56 | 140 | 4.8 (2.25,10.04) | < 0.001* | 5.8 (2.25,15.05) | < 0.001* |
| | Yes | 9 | 107 | 1 | | | |
| Spend outside at dusk | Yes | 60 | 184 | 4.1 (1.6, 10.69) | 0.004* | 2.0 (0.66, 6.23) | 0.216 |
| | No | 5 | 63 | 1 | | | |
| Irrigation within 500m from home | Yes | 24 | 38 | 3.2 (1.7, 5.93) | < 0.001* | 3.0 (1.38, 6.67) | 0.006* |
| | No | 41 | 209 | 1 | | | |

HCs = Health centers.

prevalence of malaria in this study is lower than the prevalence in Burkina Faso (49%) [19] and Mali (28.1%) [20]. This difference might be due to the difference in overall prevalence and burden of malaria in countries. Compared to other endemic countries in Sub-Saharan Africa, malaria prevalence in Ethiopia is relatively low; but, Burkina Faso and Mali are among the top ten countries with the highest number of malaria cases and deaths. Burkina Faso takes up (3% of the global cases and 4% of global deaths) and Mali covers (3% of the global cases and deaths, and 6% of cases in West Africa) [2]. In addition, the difference may be due to the difference in study period. For example, the current study was done from February to April (local dry season) and low malaria transmission season; whereas, the study in Burkina Faso was done during both low and high malaria transmission seasons and the study in Mali was done during high malaria transmission season (September and May).

On the other hand, the prevalence of malaria in the present study is higher than the prevalence in Pawe Hospital, northwestern Ethiopia (16.3%) [21] and North Gonder, northwestern Ethiopia (11.5%) [22]. This variation might be due to low coverage and utilization of LLINs and existence of favorable vector breeding sites like irrigation in the current study area. Since, 63% of pregnant women were not sleeping under LLINs mainly due to unavailability and this might have increased the prevalence in the current study.

*Plasmodium falciparum* mono-infection was the most predominant species with overall prevalence of 12.2%, which is in line with the study done in Pawe Hospital, northwestern Ethiopia (9.67%) [21]. But, this result is lower than the study conducted in Burkina Faso (26.9%) [19]. This variation might be due to the difference in inclusion criteria, study period and study site (type of health facility). The present study was conducted in health centers and pregnant women with severe malaria were excluded. However, Tahita and his colleagues' study was conducted in hospital and included pregnant women with severe malaria which mostly occurs due to *P. falciparum*. In addition, the overall malaria burden is higher in Burkina Faso compared to Ethiopia as explained above.

The prevalence of *P. vivax* in the current study was 4.8% which is in line with the study in North Gonder, northwestern Ethiopia (3%) [22]. However, this result is higher than the study done in Pawe hospital (1.6%) [21]. This variation might be due to variation in vector competence, rain fall, temperature and study site. The prevalence of mixed infections in the current study was 3.8%. This is in line with studies in North Gonder, northwestern Ethiopia (2.3%) [22] and Pawe hospital, northwestern Ethiopia (3.03%) [21].

Analysis of factors associated with malaria among symptomatic pregnant women showed that, educational status, gestational age, gravidity, distance, LLINs and irrigation were significantly associated with malaria. In the current study, educational status of pregnant women has significant association with malaria. Illiterate pregnant women had increased malaria susceptibility than women who attended secondary school and higher education. A similar finding was reported from studies done in Nigeria [23] and Benishangul Gumuz, northwest Ethiopia [24].

Pregnant women in the first trimester of pregnancy were at increased odds of having malaria than women in third trimester. This finding is in line with the study in Mali [20], Jawi district, northwestern Ethiopia [13] and Benishangul Gumuz, northwest Ethiopia [24]. In relation to gravidity, primigravidae were at higher odds of developing symptomatic malaria compared to multi-gravidae. Similar association was found in studies done in Mozambique [25], Ghana [18], Burkina Faso [19], Jawi district, northwestern Ethiopia [13] and Benishangul Gumuz, northwestern Ethiopia [24].

Distance, where pregnant women live relative to the health centers was statistically associated with symptomatic malaria infection. However, distance in relation to malaria was not assessed in any studies. According to the present study, pregnant women who live in far

distances were 2.7 times more likely to have increased malaria susceptibility compared to pregnant women living in nearby health centers. This might be due to the assumption that, women of far areas may not attend health centers regularly for their ANC follow up because of transportation problem. In addition, the health centers are situated in the towns and as we go far, there might be increased vector breeding sites that in turn might have made the association.

In the present study, pregnant women who do not sleep under LLINs have significantly increased susceptibility to malaria infection. This finding is in agreement with the study done in Benishangul Gumuz, northwest Ethiopia [24]. However, according to the studies done in Burkina Faso [19] and Jawi district, northwestern Ethiopia [13], the association between bed net utilization and malaria was not significant. This variation might be due to higher coverage and utilization of LLINs in the above studies compared to the present study (72% versus 37%, respectively).

According to the present study, the presence of irrigational activity in nearby houses was significantly associated with prevalence of malaria. Pregnant women who live in areas where irrigational activities are practiced within 500 meters in nearby houses were 3 times more likely to have malaria than those living in irrigation free areas. This might be due to the fact that, irrigation might serve as important factors for *Anopheles* mosquito breeding and might make the association significant. However, other studies have not investigated the association between irrigation and malaria.

## Conclusions

Even though prevalence of malaria is decreasing in the country because of scale-up of intervention and prevention measures, this study showed that, malaria is still the major public health problem among pregnant women. *P. falciparum* was the predominant species causing malaria infection in the area. Illiteracy, first trimester, primigravidae, living far from health centers, not sleeping under LLINs and living in areas within 500m near irrigation were factors that increased the prevalence of malaria infection in malaria symptomatic pregnant women. Therefore, special attention should be given to pregnant women prone to these factors.

## Supporting information

**S1 File.**
(ZIP)

## Acknowledgments

We would like to acknowledge College of Medicine and Health Sciences, Bahir Dar University Institutional Review Board for giving ethical approval. We would also like to express our appreciation to the study participants.

## Author Contributions

**Conceptualization:** Andargachew Almaw.

**Writing – original draft:** Andargachew Almaw.

**Writing – review & editing:** Andargachew Almaw, Mulat Yimer, Megbaru Alemu, Banchamlak Tegegne.

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
