## [Decision Letter · Decision Letter 0]

14 Dec 2021

PONE-D-21-30318Prevalence of malaria and associated factors among symptomatic pregnant women attending antenatal care at three health centers in North-west EthiopiaPLOS ONE

Dear Dr.  Almaw,

Thank you for submitting your manuscript to PLoS ONE. After careful consideration, we felt that your manuscript requires revision, following which it can possibly be reconsidered. Although your manuscript was of interest to the reviewers, major concerns were related to study design and  results.  According to the reviewers, the methods were not described in enough details to allow suitably skilled investigators to fully replicate and evaluate the study;  for example, how was defined women of interest, how about the questionnaire (demographic, clinical and socioeconomic data).   In addition, a significant number of issues should be clarified and/or adjust otherwise the MS’s results may be compromised. For your guidance, a copy of the reviewers' comments was included below

We look forward to receiving your revised manuscript.

Kind regards,

Luzia Helena Carvalho, Ph.D.

Academic Editor

PLOS ONE

Journal Requirements:

2. Please provide additional details regarding participant consent. In the ethics statement in the Methods and online submission information, please describe how verbal consent was documented and witnessed, and why written consent was not obtained. If your study included minors, state whether you obtained consent from parents or guardians.

3. Please include additional information regarding the survey or questionnaire used in the study and ensure that you have provided sufficient details that others could replicate the analyses. For instance, if you developed the survey or questionnaire as part of this study and it is not under a copyright more restrictive than CC-BY, please include a copy, in both the original language and English, as Supporting Information. If the questionnaire is published, please provide a citation to the (1) questionnaire and/or (2) original publication associated with the questionnaire.

6. We note that you have referenced (ie. Bewick et al. [5]) which has currently not yet been accepted for publication. Please remove this from your References and amend this to state in the body of your manuscript: (ie “Bewick et al. [Unpublished]”) as detailed online in our guide for authors

7. We note that Figure in your submission contain [map/satellite] images which may be copyrighted. All PLOS content is published under the Creative Commons Attribution License (CC BY 4.0), which means that the manuscript, images, and Supporting Information files will be freely available online, and any third party is permitted to access, download, copy, distribute, and use these materials in any way, even commercially, with proper attribution. For these reasons, we cannot publish previously copyrighted maps or satellite images created using proprietary data, such as Google software (Google Maps, Street View, and Earth). For more information, see our copyright guidelines: http://journals.plos.org/plosone/s/licenses-and-copyright.

a. You may seek permission from the original copyright holder of Figure to publish the content specifically under the CC BY 4.0 license.  

Reviewers' comments:

Reviewer's Responses to Questions

**Comments to the Author**

1. Is the manuscript technically sound, and do the data support the conclusions?

Reviewer #1: Yes

Reviewer #2: Yes

2. Has the statistical analysis been performed appropriately and rigorously? 

Reviewer #1: Yes

Reviewer #2: Yes

3. Have the authors made all data underlying the findings in their manuscript fully available?

Reviewer #1: Yes

Reviewer #2: Yes

4. Is the manuscript presented in an intelligible fashion and written in standard English?

Reviewer #1: No

Reviewer #2: No

5. Review Comments to the Author

Reviewer #1: Major comments

1. The information flow in the background of the main body is not adequate. Try to arrange in such a way that paragraphs connect and also within the paragraphs there is a connection between sentences. The authors may start with malaria epidemiology, then biology of the parasite, clinical presentation, etc.

2. Grammatical errors should be corrected.

Minor Comment

Line 15: Rephrase the sentence to increase clarity. I suggest the sentence read, "Malaria is a disease caused by Plasmodium species…….

Line 17: Symptomatic pregnant women……however, it is not clear these women are symptomatic for what infection/condition. Clarify.

Line 18: Symptomatic pregnant women are highly potential transmitters…..but symptomatic individuals normally seek medical attention, thus it is not clear how can they become an important cause of epidemics. It would be clear if they were asymptomatic.

Line 27: Questionnaire data was collected…..this sentence is not clear, re-write it to increase clarity. For instance, “A questionnaire was used to collect information/data through a face-to-face interview.

Line 28: Thin and thick films from…..re-write to read…”Capillary blood samples were collected and used to prepare thick and thin blood smears, which were then stained using ……”

Line 30: Simply write….”Logistic regression was used to assess……..”

Line 36: Pregnant women who couldn’t read………just write…”Illiterate pregnant women…..”

Line 38-39: Change “lived far from”…to…. “living far from”, “didn’t sleep under”…..to…”not sleeping under”, “lived near”…to…”living near”.

Line 40: I presume these were factors associated with confirmed malaria infection, therefore, should not be written as odds of symptomatic malaria but rather of malaria infection in malaria symptomatic pregnant women.

Lines 42-45: The conclusion does not include the contribution of the socioeconomic factors on the presence of confirmed malaria infection in these malaria symptomatic pregnant women.

Line 50: In which year were the 229 million cases and 409,000 deaths were reported, and where is the reference?

Line 64: RDTs should be written in a long form since it appears for the first time.

Line 53-57: During feeding……Then parasites infect and multiply…..This section is redundant, should be removed.

Methods

Study design

Line 101: The climate is Woyna Dega………can you explain what does this suppose to mean in climatic terms?

Line 104: Rephrase the sentence to read, “Mean annual rainfall and temperature of……………is 1800 mm and 20.10C, respectively.”

Line 128-29: Using proportionate allocation 87, 24, and 201 study participants were involved……where were these proportions involved from each of the three study sites, respectively? If so please state that, and explain what was the criteria for having such proportions per site.

Data collection

Line 141: Questionnaire…..can you explain whether the questions were open or close-ended? Please also give a summary of the information the questionnaire was trying to gather, I mean the type of demographic and socioeconomic or clinical data.

Line 143: Blood sample collection….Of the blood slides (thin/thick), which one was used for the detection and counting of the parasite density, and which one was used for species identification?

Did you count the parasite density? If yes, how was it done?

Results

Line 198: The prevalence of malaria………..the percentages should be out of the brackets, and the whole numbers in brackets and given as proportions with the denominator being the total number of malaria infections e.g. 44.6% (29/65), etc.

Line 198-201: This information should be given in a form of a Table, and preferably be added to Table 2. Table 2 needs to be restructured by removing the parasite stages prevalence i.e. Gametocytes and schizonts as they add no value to the data presented. Likewise, remove the number of malaria negative results and instead provide the proportions of malaria positives and their percentages e.g. P. falciparum 12.2% (38/312), P. vivax 4.8% (15/312), and mixed infection…..

Line 204-205: Provide the odds ratio, confidence interval, and p-value.

Discussion

Line 257-60: Can you explain why primigravidae had higher odds of having malaria infection than other multigravidas?

Conclusion

Should be summarized and made more clear.

Reviewer #2: Dr Almaw and colleagues have examined symptomatic pregnant women for malaria and describe the associated characteristics of the women found to have malaria. The study is quite interesting, and the results clearly identify certain groups of symptomatic pregnant women as being at particular risk of malaria. There are some missing pieces of information that will strengthen the paper.

Major comments

1. The authors need to provide a clear definition of “symptomatic”. How did they define women of interest? Also, how did they identify “pregnant”? Were pregnancy tests done in women of childbearing age, for example? This is important for interpreting relative proportions of infection by trimester. (It would also be helpful to define cutoffs used for different trimesters).

2. The manuscript contains unnecessary detail in places. For example, lines 52-60 provide a “textbook” description of the parasite life cycle, which can all be removed, and similarly lines 64-73 contain much generic information on malaria diagnosis and management in general and in pregnancy. The detailed description of the study sites can be condensed substantially (lines 97-119)- but please include the size of populations in the catchments here.

3. Sample size, Line 123. These determinations are often hard to follow. It seems the researchers assumed a parasite prevalence of 50% which was high, and was not observed, and the sample size was not obtained. How does this affect study interpretation?

4. Line 198: these data are relative proportions by trimester, not prevalence.

5. Line 216-7: This statement is manifestly incorrect and should be removed. Even if it is the first study of a very specific question such assertions are best avoided.

6. In the discussion it is generally unclear whether studies from elsewhere used similar recruitment criteria or not. This is critically important for comparing the studies. For example reference 7 appears to be a simple cross sectional study, while reference 21 (Tagbor) compared symptoms in parasitemic and aparasitemic women. References by Tahita (22) and Bardaji seem most directly comparable in design.

7. Line 219-220: differences in season relative to transmission of malaria is a possible explanation (and worth mentioning) but it would make sense to start with differences in overall transmission between countries. Isn’t disease burden much higher in the mentioned countries?

8. Line 224-5: what was the design of the other studies in Ethiopia, if cross sectional or observational rather than focussed on symptomatic women this needs to be taken into account.

9. Line 238-242: this paragraph is confusing in its intentions, as to whether the findings are similar to the two previous studies, or different than them. Keeping the comparison simple and not trying to explain “differences” is easier to follow, especially given the size of one of the earlier studies.

Minor comments

1. There are many minor English language errors throughout the manuscript. An edit by an expert English speaker or professional editor is needed. I have confined my suggestions to the scientific content.

2. Abstract and conclusion: the term “considerably high” is poor, please change it.

3. Check use of abbreviations and only use them if repeated 3 or more times. IRS, LSM, and probably LLINs are examples of unnecessary abbreviations.

4. Line 81: the figure of 11 million women exposed to malaria refers to infected pregnant women in Africa, please revise.

5. Line 133 how is “irrigation” defined?

6. Line 158-160: how was the QC implemented? What happened in case of disagreements?

7. Table 1: the lines for “total” are really superfluous and could be removed.

8. Table 3 “Spent outside at dusk” needs better description and definition.

9. Line 288-290: are these symptoms authors used to define malaria?

10. The following references need additional detail or correction: 1, 4 (journal name spelled out); 2 and 12: 2 different editions of World Malaria Report- both needed?; 9: second author surname missing; 18, 25: no source given- journal or other publication??; 21: incomplete citation- published?

6. PLOS authors have the option to publish the peer review history of their article (what does this mean?). If published, this will include your full peer review and any attached files.

Reviewer #1: **Yes: **Richard Mwaiswelo

Reviewer #2: No

---

## [Author Response · Author response to Decision Letter 0]

14 Jan 2022

Point by point response to editor and reviewers comments

1. Point by point response to editor comments

First, we authors would like to thank editor and both reviewers for comments to make this manuscript more plausible. As much as possible, we authors would like to address your comments here after.

Response: well taken. Dear editor, we have now added requirements that were previously missed. 

2. Please provide additional details regarding participant consent. In the ethics statement in the Methods and online submission information, please describe how verbal consent was documented and witnessed, and why written consent was not obtained. If your study included minors, state whether you obtained consent from parents or guardians.

Response: well taken. Dear editor, our study subjects were pregnant women who came to health center for Ante Natal Care visit seeking diagnosis and treatment for malaria. They are sick and do not refuse to give data like history, signs and/or symptoms, blood sample etc. But, to participate in the study, we explained the purpose of the study and requested their willingness without signing on consent form. This was what we want to mean by Verbal consent. 

3. Please include additional information regarding the survey or questionnaire used in the study and ensure that you have provided sufficient details that others could replicate the analyses. For instance, if you developed the survey or questionnaire as part of this study and it is not under a copyright more restrictive than CC-BY, please include a copy, in both the original language and English, as Supporting Information. If the questionnaire is published, please provide a citation to the (1) questionnaire and/or (2) original publication associated with the questionnaire.

Response: well taken. We have included as supportive file.

Response: well taken. We declare that, the authors received no specific funding for this work.

Response: Well taken. dear reviewer, all data are available without restrictions

6. We note that you have referenced (ie. Bewick et al. [5]) which has currently not yet been accepted for publication. Please remove this from your References and amend this to state in the body of your manuscript: (ie “Bewick et al. [Unpublished]”) as detailed online in our guide for authors

Response: well taken and removed

7. We note that Figure in your submission contain [map/satellite] images which may be copyrighted. All PLOS content is published under the Creative Commons Attribution License (CC BY 4.0), which means that the manuscript, images, and Supporting Information files will be freely available online, and any third party is permitted to access, download, copy, distribute, and use these materials in any way, even commercially, with proper attribution. For these reasons, we cannot publish previously copyrighted maps or satellite images created using proprietary data, such as Google software (Google Maps, Street View, and Earth)…………..

Response: dear editor, we have removed it. 

 Thank you

2. Point by point response to reviewer#1 comments

Authors’ Response to the reviewer#1`s comments

First, we authors would like to thank reviewer for comments to make this manuscript more plausible. As much as possible, we authors would like to address your comments here after.

Reviewer #1: Comments to the author

Reviewer #1: Major comments

1. The information flow in the background of the main body is not adequate. Try to arrange in such a way that paragraphs connect and also within the paragraphs there is a connection between sentences. The authors may start with malaria epidemiology, then biology of the parasite, clinical presentation, etc.

Response: well taken and corrected. 

2. Grammatical errors should be corrected.

Response: well taken. We have tried to edit

Minor Comments

Line 15: Rephrase the sentence to increase clarity. I suggest the sentence read, "Malaria is a disease caused by Plasmodium species…….

Response: Well taken and we have corrected as you suggested

Line 17: Symptomatic pregnant women……however, it is not clear these women are symptomatic for what infection/condition. Clarify.

Response: well taken and it is to mean malaria symptomatic

Line 18: Symptomatic pregnant women are highly potential transmitters…..but symptomatic individuals normally seek medical attention, thus it is not clear how can they become an important cause of epidemics. It would be clear if they were asymptomatic.

Response: Well taken. Dear reviewer, Symptomatic pregnant women have higher potential of attracting mosquitoes because of different physiological changes like increased abdominal temperature and hence sweating, increased breath etc. so that they can be easily bitten and that vector again may take infection and transmit to the others.

Line 27: Questionnaire data was collected…..this sentence is not clear, re-write it to increase clarity. For instance, “A questionnaire was used to collect information/data through a face-to-face interview.

Response: Well taken and we have corrected as you suggested

Line 30: Simply write….”Logistic regression was used to assess……..”

Response: well taken and corrected

 Line 36: Pregnant women who couldn’t read………just write…”Illiterate pregnant women…..”

Response: well taken and corrected

Line 38-39: Change “lived far from”…to…. “living far from”, “didn’t sleep under”…..to…”not sleeping under”, “lived near”…to…”living near”.

Response: well taken and changed

Line 40: I presume these were factors associated with confirmed malaria infection, therefore, should not be written as odds of symptomatic malaria but rather of malaria infection in malaria symptomatic pregnant women.

Response: well taken and corrected

Lines 42-45: The conclusion does not include the contribution of the socioeconomic factors on the presence of confirmed malaria infection in these malaria symptomatic pregnant women.

Response: well taken and corrected

Line 50: In which year were the 229 million cases and 409,000 deaths were reported, and where is the reference?

Response: well taken; it was in 2019 and the reference is WHO 2020

Line 64: RDTs should be written in a long form since it appears for the first time.

Response: well taken and corrected

Line 53-57: During feeding……Then parasites infect and multiply…..This section is redundant, should be removed.

Response: Well taken 

Methods

Study design

Line 101: The climate is Woyna Dega………can you explain what does this suppose to mean in climatic terms?

Response: well taken. Dear reviewer, Woyna dega (Amharic term) is climatic zone covering areas of 1500m -2500m

Line 104: Rephrase the sentence to read, “Mean annual rainfall and temperature of……………is 1800 mm and 20.10C, respectively.”

Response: well taken and rephrased

Line 128-29: Using proportionate allocation 87, 24, and 201 study participants were involved……where were these proportions involved from each of the three study sites, respectively? If so please state that, and explain what was the criteria for having such proportions per site.

Response: well taken. Dear reviewer, it is to mean that; 87, 24, and 201 study participants from Tis Abay, Zenzelma and Hamusit health centers, respectively. This proportions per site were obtained by taking previously diagnosed malaria symptomatic pregnant women from each health center, i.e 336, 90 and 776 ( line 118 previous document) malaria symptomatic pregnant women from Tis Abay, Zenzelma and Hamusit health centers, respectively were diagnosed in precious year( 2019/2020). Total (336+90+776 = 1202). So, if there are 336 (for Tis Abay health center) from 1202, how many will be selected from 312(sample size)? It is equal to 76. Using similar way for getting number of malaria symptomatic pregnant women to be enrolled from each health center; i.e 24 and 201 for Zenzelma and Hamusit health centers, respectively.

Data collection

Line 141: Questionnaire…..can you explain whether the questions were open or close-ended? Please also give a summary of the information the questionnaire was trying to gather, I mean the type of demographic and socioeconomic or clinical data.

Response: dear reviewer, questionnaire was closed ended. The socio demographic data we gathered include; age, educational status, occupation, residence, etc. the clinical data were signs and/or symptoms of malaria. For more information, can you please see the questionnaire we have attached as supportive file? All the data we gathered are available there. 

Line 143: Blood sample collection….Of the blood slides (thin/thick), which one was used for the detection and counting of the parasite density, and which one was used for species identification?

Did you count the parasite density? If yes, how was it done?

Response: Thick blood films were used for detection and counting of parasite density; whereas, thin blood films were used for species identification. Malaria parasite count was performed on Giemsa-stained thick blood film against 200 WBCs. Malaria parasite density was calculated based on the assumption of 8,000 WBC per micro-liter (µl) of blood.

Parasites/ µl = Parasite counted against 200 WBCs×8000 WBCs/ µl 

 200

Results

Line 198: The prevalence of malaria………..the percentages should be out of the brackets, and the whole numbers in brackets and given as proportions with the denominator being the total number of malaria infections e.g. 44.6% (29/65), etc.

Response: well taken and corrected

Line 198-201: This information should be given in a form of a Table, and preferably be added to Table 2. Table 2 needs to be restructured by removing the parasite stages prevalence i.e. Gametocytes and schizonts as they add no value to the data presented. Likewise, remove the number of malaria negative results and instead provide the proportions of malaria positives and their percentages e.g. P. falciparum 12.2% (38/312), P. vivax 4.8% (15/312), and mixed infection…..

Response: well taken; dear reviewer, we have corrected as you suggested. Please see pages 11 line 186-187

Line 204-205: Provide the odds ratio, confidence interval, and p-value.

Response: Dear reviewer, do you mean from the table? If so, there is COR, AOR and p-value.

Discussion

Line 257-60: Can you explain why primigravidae had higher odds of having malaria infection than other multigravidas?

Response: dear reviewer, this may be related to the development of pre-immunity and antibodies in multigravidae. So, the development of pre-immunity to malaria with increased gravidity and previous exposures may help multigravidae women to be in reduced risk of malaria. But, primigravidae lack this pre-immunity. It might be also linked to infection-specific immunological factors. Plasmodium-infected erythrocytes sequester in the maternal placenta by producing surface antigens mainly variant surface antigens (VSA) that bind to chondroitin Sulphate-A (CSA) receptors in the placenta. These antibodies are associated with protection against placental infection. Therefore, primigravidae mothers lack these anti-adhesion antibodies against CSA binding parasites, which develop only after successive pregnancies and this makes them more susceptible to infection (Fried et al., 1998).

Conclusion

Should be summarized and made more clear

Response: well taken: we have summarized 

 Thank you

3. Point by point response to reviewer#2 comments

Authors’ Response to the reviewer`s comments

First, we authors would like to thank reviewer#2` for comments to make this manuscript more plausible. As much as possible, we authors would like to address your comments here. 

 Reviewer #2: Dr Almaw and colleagues have examined symptomatic pregnant women for malaria and describe the associated characteristics of the women found to have malaria. The study is quite interesting, and the results clearly identify certain groups of symptomatic pregnant women as being at particular risk of malaria. There are some missing pieces of information that will strengthen the paper.

Major comments

1. The authors need to provide a clear definition of “symptomatic”. How did they define women of interest? Also, how did they identify “pregnant”? Were pregnancy tests done in women of childbearing age, for example? This is important for interpreting relative proportions of infection by trimester. (It would also be helpful to define cutoffs used for different trimesters)

Response: well taken. We defined “symptomatic” as Women who exhibited at least two of the following signs and /or symptoms of malaria;

Fever (temperature ≥37.5c0 as recorded by electronic thermometer), joint pain, headache, vomiting, malaise, chills…

Pregnant: women of confirmed urine HCG positive

2. The manuscript contains unnecessary detail in places. For example, lines 52-60 provide a “textbook” description of the parasite life cycle, which can all be removed, and similarly lines 64-73 contain much generic information on malaria diagnosis and management in general and in pregnancy. The detailed description of the study sites can be condensed substantially (lines 97-119) but please include the size of populations in the catchments here.

Response: well taken. Dear reviewer, we have edited study area and included number of catchment populations getting service from each health center. Please see page 6. But, we haven’t omitted life cycle, diagnosis and management section. Because, the second reviewer suggested that, this section is in adequate and should even be added. We also believe that, this is relevant in back ground section. We hope that, you will suggest as again and we authors will accept and re-write.

Sample size, Line 123. These determinations are often hard to follow. It seems the researchers assumed a parasite prevalence of 50% which was high, and was not observed, and the sample size was not obtained. How does this affect study interpretation? 

Response: dear reviewer, we were unable to address the sample size obtained by taking 50% because of logistic reasons (like time and other resources). But, we have reduced it according to statistical reasons. 

4. Line 198: these data are relative proportions by trimester, not prevalence.

Response: well taken. Dear author, we have corrected as you suggested. 

5. Line 216-7: This statement is manifestly incorrect and should be removed. Even if it is the first study of a very specific question such assertions are best avoided.

Response: well taken and corrected

6. In the discussion it is generally unclear whether studies from elsewhere used similar recruitment criteria or not. This is critically important for comparing the studies. For example reference 7 appears to be a simple cross sectional study, while reference 21 (Tagbor) compared symptoms in parasitemic and aparasitemic women. References by Tahita (22) and Bardaji seem most directly comparable in design.

Response: well taken. Dear reviewer, it is true that some studies we used do not have similar study design as well as recruitment criteria and even the objectives are different. We simply took the overall and species specific prevalence and compared. Because; as our understanding, we couldn’t get studies which are closely related to our study by study design, recruitment criteria, objective… So, to discuss the prevalence, we have used those studies for comparison.

7. Line 219-220: differences in season relative to transmission of malaria is a possible explanation (and worth mentioning) but it would make sense to start with differences in overall transmission between countries. Isn’t disease burden much higher in the mentioned countries?

Response: Alright! Overall, Burkina Faso and Mali are among the ten countries with the highest number of malaria cases and deaths. Burkina Faso takes up (3% of the global cases and 4% of global deaths) and Mali covers (3% of the global cases and deaths, and 6% of cases in West Africa) (WHO 2020). Compared to other endemic countries in Sub-Saharan Africa, malaria prevalence in Ethiopia is relatively low.

Line 224-5: what was the design of the other studies in Ethiopia, if cross sectional or observational rather than focused on symptomatic women this needs to be taken into account.

Response: Well taken. It was similar study design (cross-sectional).

9. Line 238-242: this paragraph is confusing in its intentions, as to whether the findings are similar to the two previous studies, or different than them. Keeping the comparison simple and not trying to explain “differences” is easier to follow, especially given the size of one of the earlier studies.

Response: well taken. Dear reviewer, We have removed this part as we found it not important 

Minor comments

1. There are many minor English language errors throughout the manuscript. An edit by an expert English speaker or professional editor is needed. I have confined my suggestions to the scientific content.

Response: well taken and we have tried to edit

2. Abstract and conclusion: the term “considerably high” is poor, please change it.

Response: well taken; dear reviewer, we have changed it. Please see in page 3

3. Check use of abbreviations and only use them if repeated 3 or more times. IRS, LSM, and probably LLINs are examples of unnecessary abbreviations.

Response: well taken and accepted 

4. Line 81: the figure of 11 million women exposed to malaria refers to infected pregnant women in Africa, please revise.

Response: well taken and we have corrected

5. Line 133 how is “irrigation” defined?

Response: well taken; irrigation in our study terms is defined as “agricultural activity whereby farmers use controlled amount of water through canals for watering and cultivation of rice, onion and sugar cane.”

6. Line 158-160: how was the QC implemented? What happened in case of disagreements?

Response: well taken. Dear reviewer, the main goal of examining the slides by malaria microscopists in APHI was to confirm results and give final decision. So that, the disagreements were all confirmed and decided by those malaria microscopists. 

7. Table 1: the lines for “total” are really superfluous and could be removed.

Response: well taken and removed 

8. Table 3 “Spent outside at dusk” needs better description and definition.

Response: well taken. Dear reviewer, there are people who do not be indoor at dusk/evening/ especially in rural areas in Ethiopia. They spend most of the time in evening performing different activities outside. They may even sit together and play with family members and neighbors. Because, this is the only time they can have a rest after working whole days in farm. They may be bitten by mosquitoes during the time. This is what we want to mean by “spend outside at dusk”

9. Line 288-290: are these symptoms authors used to define malaria?

Response: well taken. Of course! But, it seems that this should not be put here and we removed.

10. The following references need additional detail or correction: 1, 4 (journal name spelled out); 2 and 12: 2 different editions of World Malaria Report- both needed?; 9: second author surname missing; 18, 25: no source given- journal or other publication??; 21: incomplete citation- published

Response: well taken. References 1&4 are corrected.

*We have used two different editions of WHO. Because, we have used different data from both editions 

* Reference 9; well taken and corrected as Hyiid L,

Reference 18: well taken and we have removed as it is un-necessary

Reference 21 from previous manuscript (reference19 from revised manuscript after reviewers’ comments); dear reviewer, we are un clear with mistakes from this reference. Can you please suggest us? Reference 21(reference19 from revised manuscript); ??? Please see page20, line 333-335

Reference 25 (ref 23 from revised manuscript, page 21): has been corrected

 Thank you

---

## [Decision Letter · Decision Letter 1]

1 Feb 2022

PONE-D-21-30318R1Prevalence of malaria and associated factors among symptomatic pregnant women attending antenatal care at three health centers in North-west EthiopiaPLOS ONE

Dear Dr. Almaw,

Thank you for submitting your manuscript to PLoS ONE. After careful consideration, we feel that your manuscript will likely be suitable for publication if the authors revise it to address specific points raised now by the reviewer. According to the reviewer, there are some specific areas where further improvements would be of substantial benefit to the readers.   For your guidance, a copy of the reviewers' comments was included below. 

We look forward to receiving your revised manuscript.

Kind regards,

Luzia Helena Carvalho, Ph.D.

Academic Editor

PLOS ONE

Journal Requirements:

Reviewers' comments:

Reviewer's Responses to Questions

**Comments to the Author**

1. If the authors have adequately addressed your comments raised in a previous round of review and you feel that this manuscript is now acceptable for publication, you may indicate that here to bypass the “Comments to the Author” section, enter your conflict of interest statement in the “Confidential to Editor” section, and submit your "Accept" recommendation.

Reviewer #1: All comments have been addressed

Reviewer #2: (No Response)

2. Is the manuscript technically sound, and do the data support the conclusions?

Reviewer #1: Yes

Reviewer #2: Yes

3. Has the statistical analysis been performed appropriately and rigorously? 

Reviewer #1: Yes

Reviewer #2: Yes

4. Have the authors made all data underlying the findings in their manuscript fully available?

Reviewer #1: No

Reviewer #2: Yes

5. Is the manuscript presented in an intelligible fashion and written in standard English?

Reviewer #1: Yes

Reviewer #2: Yes

6. Review Comments to the Author

Reviewer #1: (No Response)

Reviewer #2: The authors have largely responded to my suggestions, and the requested details have mostly been provided.

Minor comments

1. Line 48: the paper shows that malaria is A significant problem, not THE significant problem.

2. I still contend that lines 63-71 are text book descriptions of malaria which do not belong in this article. Reviewer 1 suggests also that some of this text is redundant.

3. Line 136 Sample size: I still do not find this clear, and the authors’ written response is also unclear (response to reviewers, page 11).

4. Line 185-6: The wording is not clear here. If the expert microscopist’s review was taken as final, please say so. Or describe who confirmed discordant results and how.

7. PLOS authors have the option to publish the peer review history of their article (what does this mean?). If published, this will include your full peer review and any attached files.

Reviewer #1: No

Reviewer #2: **Yes: **Stephen Rogerson

---

## [Author Response · Author response to Decision Letter 1]

8 Mar 2022

Point by point response to reviewer#2 comments

First of all, we authors would like to thank reviewer#2` again for comments to make this manuscript more plausible. As much as possible, we authors would like to address your comments here.

Minor comments

1. Line 48: the paper shows that malaria is A significant problem, not THE significant problem.

Response: well taken; dear reviewer, we have corrected as you suggested. “Malaria is the disease…”

2. I still contend that lines 63-71 are text book descriptions of malaria which do not belong in this article. Reviewer 1 suggests also that some of this text is redundant.

Response: well taken; dear reviewer, this section is about the transmission and life cycle in human host. Actually, it is in lines 49-55 from revised manuscript. We understood that the idea was… “Malaria is transmitted to humans by the bite of female Anopheles mosquitoes. Anopheles mosquitoes feed on blood for survival and production of eggs. During feeding, infected female Anopheles mosquito inoculates the infective sporozoites from its salivary gland into human skin. After inoculation, parasites circulate with blood and those reaching to the liver undergo one cycle development. Then, parasites infect and multiply inside red blood cells to bring the characteristic signs and symptoms. The clinical manifestations include fever, joint pain, chills, headache and vomiting which usually appear between 10 and 15 days after the Anopheles mosquito bite. In addition, malaria may also be transmitted through blood transfusion and congenitally”. Even we haven’t got redundancy here. We hope we have got your idea. Thank you!

3. Line 136 Sample size: I still do not find this clear, and the authors’ written response is also unclear (response to reviewers, page 11).

Response: well taken; dear reviewer, 

The sample size was determined using single population proportion formula based on 50% prevalence, 95% confidence level and 5% margin of error with 10% non-response rate as follows.

n = (Zα/2)2 P (1-P) 

 d2 

n= (1.96)2 (0.5) (1–0.5)2 = 384 plus 10% non-response rate =422 

 (0.05)2

Where;

n = number of sample size.

P= the proportion (p = 0.5) taken as 50% prevalence 

d= marginal error between the sample and population (0.05). 

Z= critical value at 95% certainty (1.96), considering 10 % non-responsive rate. 

But, according to information obtained from health centers, the total number of symptomatic pregnant women diagnosed in the three study areas in 2019/2020 was 1202 which is finite population or less than 10.000. So, according to statistics, adjustment (correction) formula is used to calculate sample sizes for population of less than 10,000. This is because a given sample size provides proportionately more information for small population than for a large population and we have reduced the sample size based on the following formula. 

nf = n/(1+(n/N)) where, nf= final sample size i.e. sample size from finite population

 n= sample size from an infinite population i.e. 422

 N= total number of target population from all three health centers who visited in previous year i.e. 1202; So, nf= 422/1+ (422/1202)) = 312 (total number of symptomatic pregnant women participated in the study).

4. Line 185-6: The wording is not clear here. If the expert microscopist’s review was taken as final, please say so. Or describe who confirmed discordant results and how.

Response: Well taken. Dear reviewer, the review (re-examination) by microscopists in APHI was taken as final. Because, it is the APHI that assesses malaria quality assurance and gives final decision about species type in every health institutions around. 

 Thank you!

---

## [Decision Letter · Decision Letter 2]

18 Mar 2022

PONE-D-21-30318R2Prevalence of malaria and associated factors among symptomatic pregnant women attending antenatal care at three health centers in North-west EthiopiaPLOS ONE

Dear Dr. Almaw,

Thank you for submitting your manuscript for review to PLoS ONE. After careful consideration, we feel that your manuscript will likely be suitable for publication if the authors revise it to address critical points raised now by the reviewers.  According to reviewers, there are some specific areas where further improvements would be of substantial benefit to the readers.A copy of the reviewers’ comments was included for your information.  

We look forward to receiving your revised manuscript.

Kind regards,

Luzia Helena Carvalho, Ph.D.

Academic Editor

PLOS ONE

Journal Requirements:

Reviewers' comments:

Reviewer's Responses to Questions

**Comments to the Author**

1. If the authors have adequately addressed your comments raised in a previous round of review and you feel that this manuscript is now acceptable for publication, you may indicate that here to bypass the “Comments to the Author” section, enter your conflict of interest statement in the “Confidential to Editor” section, and submit your "Accept" recommendation.

Reviewer #1: All comments have been addressed

Reviewer #2: All comments have been addressed

2. Is the manuscript technically sound, and do the data support the conclusions?

Reviewer #1: Yes

Reviewer #2: Yes

3. Has the statistical analysis been performed appropriately and rigorously? 

Reviewer #1: Yes

Reviewer #2: Yes

4. Have the authors made all data underlying the findings in their manuscript fully available?

Reviewer #1: Yes

Reviewer #2: Yes

5. Is the manuscript presented in an intelligible fashion and written in standard English?

Reviewer #1: No

Reviewer #2: Yes

6. Review Comments to the Author

Reviewer #1: Abstract

Background

Line 18......There is no need of having this sentence since the previous sentence in line 15-17 has already explained the problem that prompted this study to be conducted.

Results

There should be no space between the numbers and the percentage symbol, but rather there should be a space between the percentage symbol and the brackets. The correction should be done throughout the document.

Conclusion

Lines 38-40..should state that...."malaria is still a public health problem among pregnant women" ....instead of ..."in the study area"

Line 42....change "are" to "were", "increase" to "increased/increasing" the prevalence........

Main body

Line 46.....the sentence is left hanging, here it would be good to mention the five Plasmodium species infecting human.

52.....Does the Anopheles inoculate the infective sporozoites into the skin?

Line 69-70...This sentence is not supposed to to be in that paragraph as it brings confusion. Whereas the previous sentence is on control measures, this sentence is on the risk factors for malaria infection in pregnant women.......thus there is no continuation.

Line 133.....What is HCG? Spell it out.

Line 176.....Remove the word "have".

Table 2.....Remove the frequency column as the figures in this column have already presented in the next column.

Line 189...change "have" to "had". Do the same for line 191.

Line 193....change "are" to "were"

Line 200....."this study results"...should be changed to..."the prevalence of malaria in this study".

Line 201...."than study results"....to...."than the prevalence in Burkina Faso".

Line 212...."than the study results".....to..."than the prevalence in".

Line 233....."shown"...to..."showed"

Reviewer #2: The authors have responded to my comments. I don't think textbook description of the malaria life cycle is a good use of space but will let this pass.

7. PLOS authors have the option to publish the peer review history of their article (what does this mean?). If published, this will include your full peer review and any attached files.

Reviewer #1: No

Reviewer #2: No

---

## [Author Response · Author response to Decision Letter 2]

21 Mar 2022

Point by point response to reviewers, comments

Authors’ Response to the reviewers, comments

First, we authors would like to thank reviewers` for their comments to make this manuscript more plausible. As much as possible, we authors would like to address your comments here. 

Reviewer #1: Abstract

Background

Line 18......There is no need of having this sentence since the previous sentence in line 15-17 has already explained the problem that prompted this study to be conducted.

Response: Well taken. Dear reviewer, we have removed it

Results

There should be no space between the numbers and the percentage symbol, but rather there should be a space between the percentage symbol and the brackets. The correction should be done throughout the document.

Response: Well taken. We have corrected

Conclusion

Lines 38-40.should state that...."malaria is still a public health problem among pregnant women" ....instead of ..."in the study area

Response: Well taken and we have corrected as you suggested

Line 42....change "are" to "were", "increase" to "increased/increasing" the prevalence........

Response: Well taken and changed

Main body

Line 46.....the sentence is left hanging, here it would be good to mention the five Plasmodium species infecting human.

Response: Well taken. We have included five species.

52.....Does the Anopheles inoculate the infective sporozoites into the skin?

Response: Well taken. Dear reviewer, it is to mean human circulation and we have corrected it.

Line 69-70...This sentence is not supposed to to be in that paragraph as it brings confusion. Whereas the previous sentence is on control measures, this sentence is on the risk factors for malaria infection in pregnant women.......thus there is no continuation.

Response: Well taken. We have removed it.

Line 133.....What is HCG? Spell it out.

Response: Well taken. Dear reviewer, it is to mean Human Chorionic Gonadotropin hormone and we have included it.

Line 176.....Remove the word "have".

Response: Well taken and we have removed it.

Table 2.....Remove the frequency column as the figures in this column have already presented in the next column.

Response: Well taken and we have removed.

Line 189...change "have" to "had". Do the same for line 191.

Response: Well taken. We have changed.

Line 193....change "are" to "were"

Response: Well taken. We have changed.

Line 200....."this study results"...should be changed to..."the prevalence of malaria in this study".

Response: Well taken and changed.

Line 201...."than study results"....to...."than the prevalence in Burkina Faso".

Response: Well taken and changed.

Line 212...."than the study results".....to..."than the prevalence in".

Response: Well taken and changed.

Line 233....."shown"...to..."showed"

Response: Well taken and changed

Reviewer #2: The authors have responded to my comments. I don't think textbook description of the malaria life cycle is a good use of space but will let this pass.

Response: Dear reviewer, we have received. Thank you!

---

## [Editor Report · Decision Letter 3]

22 Mar 2022

Prevalence of malaria and associated factors among symptomatic pregnant women attending antenatal care at three health centers in North-west Ethiopia

PONE-D-21-30318R3

Dear Dr. Almaw,

We’re pleased to inform you that your manuscript has been judged scientifically suitable for publication and will be formally accepted for publication once it meets all outstanding technical requirements.

Kind regards,

Luzia Helena Carvalho, Ph.D.

Academic Editor

PLOS ONE
---

## [Editor Report · Acceptance letter]

29 Mar 2022

PONE-D-21-30318R3 

Prevalence of malaria and associated factors among symptomatic pregnant women attending antenatal care at three health centers in north-west Ethiopia 

Dear Dr. Almaw:

I'm pleased to inform you that your manuscript has been deemed suitable for publication in PLOS ONE. Congratulations! Your manuscript is now with our production department. 

Kind regards, 

on behalf of

Dr. Luzia Helena Carvalho 

Academic Editor

PLOS ONE